# Role of pH Value in Clinically Relevant Diagnosis

**DOI:** 10.3390/diagnostics10020107

**Published:** 2020-02-16

**Authors:** Shu-Hua Kuo, Ching-Ju Shen, Ching-Fen Shen, Chao-Min Cheng

**Affiliations:** 1Institute of Biomedical Engineering, National Tsing Hua University, Hsinchu 300, Taiwan; mitsunari0408@gmail.com; 2College of Medicine, Kaohsiung Medical University, Kaohsiung 807, Taiwan; chenmed.tw@yahoo.com.tw; 3Department of Pediatrics, National Cheng Kung University Hospital, College of Medicine, National Cheng Kung University, Tainan 704, Taiwan

**Keywords:** pH value, diagnosis, skin, wound

## Abstract

As a highly influential physiological factor, pH may be leveraged as a tool to diagnose physiological state. It may be especially suitable for diagnosing and assessing skin structure and wound status. Multiple innovative and elegant smart wound dressings combined with either pH sensors or drug control-released carriers have been extensively studied. Increasing our understanding of the role of pH value in clinically relevant diagnostics should assist clinicians and improve personal health management in the home. In this review, we summarized a number of articles and discussed the role of pH on the skin surface as well as the factors that influence skin pH and pH-relevant skin diseases, but also the relationship of skin pH to the wound healing process, including its influence on the activity of proteases, bacterial enterotoxin, and some antibacterial agents. A great number of papers discussing physiological pH value have been published in recent decades, far too many to be included in this review. Here, we have focused on the impact of pH on wounds and skin with an emphasis on clinically relevant diagnosis toward effective treatment. We have also summarized the differences in skin structure and wound care between adults and infants, noting that infants have fragile skin and poor skin barriers, which makes them more vulnerable to skin damage and compels particular care, especially for wounds.

## 1. Introduction

Skin, the major portion of the integumentary system, is the human body’s largest organ. It spans approximately 2 m^2^ and is 0.075–0.15 mm thick in the average adult [1]. It acts as the first line of defense to shield the human body from ultraviolet radiation, infection by pathogens, and chemical irritants [2]. Furthermore, skin plays a key role in thermoregulation, immunological function, and maintaining body water balance.

Schade and Marchionini first introduced the term “acid mantle” to describe the slight acidification of the uppermost layer of human skin [3]. The acid mantle is characterized by a pH value of 4–6, as a result of amino acids, fatty acids, sebum secreted by the sebaceous gland, and lactate excreted from sweat. All of these compounds are acidic and when present together on the human skin, become a barrier that prevents bacterial colonization. Another critical skin barrier is the lipid barrier, the extracellular lipid matrix of the stratum corneum (SC) [4], which is composed of free fatty acids, cholesterol and ceramides and functions as a hydrophobic barrier for human skin [5]. A skin barrier schematic is provided in Figure 1.

Most human-pathogen bacteria are inhibited by the acidic milieu on the surface of normal human skin, but this shield is disturbed when skin is wounded. The tissue beneath skin has a physiological pH of 7.4 [6], which raises the overall pH value at the wound site and provides advantages for bacterial colonization. Bacterial contaminants in wounds, e.g., *Staphylococcus aureus*, interfere with the normal wound healing process.

There are four phases in normal wound healing [7,8]: hemostasis, inflammation, proliferation, and remodeling. During hemostasis the wound is filled with fibrin and coagulated blood. Clots are formed to stop bleeding and seal the wound site until tissues are repaired. During inflammation, histamine is produced by basophils and mast cells to increase capillary permeability, which allows leukocytes such as neutrophils to migrate to the infected wound site and remove dead cells and pathogens. However, a failure in eliminating pathogens or the remaining high pH environment can be primary reasons for chronic wound development. During proliferation, epithelial cells increase in number, granulation tissue forms, and angiogenesis occurs. Healthy granulation tissues should be pink in color and uneven in texture. The final phase, remodeling, may take years to complete. The wound matrix undergoes degradation by metalloproteinases and new extracellular matrix (ECM) is created during remodeling. Scar tissue can form due to degradation or ECM generation disruption. While the alkaline milieu activates protease, which facilitates the removal of damaged components, excessive amounts of protease eventually destroy newly constructed tissue [9]. Moreover, alkaline pH values are usually found in chronic wounds, and are associated with an increased risk of bacterial colonization [10].

pH value is a critical factor in the wound healing process and could offer important physiological condition information regarding skin status and infection. It assists physicians with clinically relevant diagnosis and patient wounds care. It can be used, for instance, to determine whether wounds are infected, becoming chronic, provide monitored, real-time wound status, and ultimately facilitate proper treatment in a more timely manner.

Recently, multiple innovative and elegant smart wound dressings combined with either pH sensors or drug control-released carriers have been extensively studied [11,12,13,14,15]. With these promising technologies, wound monitoring procedures can be visualized and designed for use in clinics and at home. The first step in clinically managing and in self-managing wounds is to understand wound status. Advances in and implementation of such technologies may alter the course of wound care and improve quality of life.

The role of pH value in evaluating skin and wound status will be systematically discussed in this review. We will emphasize its clinical application and highlight significant research that leverages pH to promote precise diagnoses leading to the development of impactful medical devices. We will also discuss pH-related differences in caring for adult skin versus neonatal skin, which is thin and requires more rigorous efforts for effective care.

## 2. Skin

As the vital, first line of defense between the body and the environment, skin is indispensable to human life. Skin structure has been thoroughly described in the literature. It is made up of three major strata, the epidermis, the dermis, and subcutaneous tissue. The epidermis, the outermost layer in skin structure, is approximately 0.05–1 mm thick, provides barrier functions, prevents infection, and regulates transepidermal water loss (TEWL). This layer is constructed of multiple sub-layers; stratum corneum, stratum lucidum, stratum granulosum, stratum spinosum, and stratum basale. The acid mantle and lipid barrier can be found on top of the epidermis, specifically just above the stratum corneum. Figure 2a is a schematic illustrating the structure of the epidermis. The second of the skin’s major strata, the dermis, can be divided into two layers, the papillary region and the reticular region. The papillary region is the thin and superficial area adjacent to the epidermis. The subsequent layer, the reticular region, is thick and contains arector pili muscles, sebaceous glands, sweat gland ducts, merocrine sweat glands, hair follicles, blood capillaries, sensory receptors, and nerve fibers. The dermis layer is approximately 1–2 mm thick, and comprises collagen, elastic fibers, and extrafibrillar matrix for providing support. The third and deepest of skin’s major three strata, subcutaneous tissue, contains fibroblasts, adipose cells, and macrophages. Subcutaneous tissue is also known as hypodermis and ranges in thickness from 1.65 mm to 18.20 mm depending on gender and skin site [16]. The hypodermis layer is used for fat storage, but it also contains large quantities of loose connective tissue and blood vessels. The structure of the dermis and hypodermis layers are depicted in Figure 2b.

### 2.1. pH Values on Skin Surfaces

The skin pH of newborns is weakly alkaline (pH of 7.4) as a result of being encapsulated within the amniotic fluid and the vernix caseosa. Hans, et al., determined and described skin pH characteristics from 209 newborn infants [17]. However, after birth, the weakly alkaline skin becomes acidic in its quest to form the acid mantle [18,19] and the functional maturation of stratum corneum is accelerated within 2 to 8 weeks [20]. During this period, infant skin remains highly prone to damage.

During sexual maturation, the pH of skin in the axillary vault climbs from around 5.0 to near 7.0, and then returns to a lower pH during sexual involution [21,22]. A. Zlotogorski [23] measured the pH distribution on the surface of the skin (forehead and cheek) of 574 men and women aged 18 to 95, and found that those over the age of 80 had higher skin surface pH values compared to the other groups, which displayed skin surface pH values between 4.0 and 5.5 on the forehead and between 4.2 and 5.9 on the cheek. There was no significant pH difference based on gender. Irvin H. Blank [24] obtained skin surface pH values (forearm, antecubital, elbow, upper arm, forehead, and back of neck) from 100 males and 100 females aged 19–27 and found pH values varying from 4.0 to 7.0, with the most frequent readings being between 4.2 and 5.6, and evidence suggesting that females have a slightly higher skin pH (0.5 higher) than males. C. Ehlers and coworkers [25] found that there was a statistically significant difference in skin pH (arm) between men and woman. Notably, measurements could not be conducted close to the wrist because most people wash their hands several times each day, and soaps can make the skin more alkaline. Another study conducted by S. Luebberding and coworkers [26] followed strict criteria and found pH differences between males and females in TEWL, SC hydration, sebum content, and at the skin surface (forehead, forearm, hand and cheek). Table 1 (adapted from [26]) displays gender-related data showing that the pH values from five skin sites and across all age groups were higher in females compared to males. This data also shows that the mean evaporimetry and corneometry values were higher for the females of this group, while the mean sebumetry values were higher for the males.

It is common to obtain inconsistent pH measurements from men and women, but additional research could provide data to support a pattern of pH value differences between genders.

### 2.2. Factors That Affects Skin pH

Regulating skin surface pH is critical to physiological defense mechanisms, but skin is affected by a number of endogenous and exogenous factors [27], including age, ethnicity, sebum, sweat, detergents, cosmetic products, soaps, occlusive dressing, and more. These factors and their influence are displayed in Table 2 [23,26,27,28,29,30,31,32,33,34,35,36,37,38,39,40,41,42,43,44,45,46,47].

Elizabeth A. and coworkers [48] analyzed the distribution of the human skin microbiome via examination of 16S ribosomal RNA gene sequences, and found that bacteria in sebaceous sites where *Propionibacterium acnes* predominated were less diverse, less even, and less rich than the bacterial milieu in moist and dry sites, which were predominantly associated with Firmicutes and Proteobacteria. Multiple factors such as sweat, sebum secretion, and pH could have correlation with predominant microbiota. This study revealed a pattern of bacterial distribution on human skin that may have implications for the future treatment of pH-relevant skin disorders, a topic that we will explore further.

### 2.3. Skin Diseases and pH Values

Beneath the acid mantle and lipid barrier, the various strata of skin have relatively static and regulated physiological pH values, but when disruptions in the protective barriers occur, or when normal skin pH is disrupted, the risk for specific skin diseases increases. Many studies have suggested a relationship between skin surface pH and disease.

It has been reported that great numbers of *Staphylococcus aureus* exist on the skin of patients with eczema can exacerbate the disease [49]. *Staphylococcus aureus* has a capacity for adhering to atopic corneocytes; this adherence phenomenon is partially attributable to the presence of protein A in its wall [50]. The pH range for optimal growth of *Staphylococcus aureus* is between 7.0 and 7.5, yet the bacteria are able to reproduce at pH values from 4.5 to 9.0. The pH value of the normal skin surface, 4.5–6.0, is consequently insufficient to kill the bacteria completely. However, the staphylococcal enterotoxin C2 (SEC2) can be destroyed in strongly acidic pH. Zinc ion, present in the crystalline structure of SEC2, may be removed completely in this process, which distorts SEC2 from its normal configuration [51].

pH on the skin surface increases in patients with seborrheic dermatitis [52,53,54]. *Pityrosporum ovule* prosper in scale of patients with dandruff, since higher pH values also promote the growth of yeast. Its colonization in less acidic areas of the skin result in declining activity of calf thymus histones, parotid saliva, lysozyme, and proteins obtained from human neutrophil granulocyte, which are cationic substances critically related to bacterial activity [55,56,57,58].

Diaper dermatitis is frequently diagnosed in infants, and wearing diapers is associated with increases in wetness and skin pH value. Ronald and coworkers [59] carried out a study involving 1601 infants. Their results suggest that diapered skin has approximately twice the TEWL of undiapered skin. Diapered skin has a higher pH value (5.9) while undiapered skin (pH 5.3) remains related to the acid mantle pH. Skin wetness could disturb the integrity of skin by increasing its permeability to irritants, frictional coefficient, and promoting microbial growth [60]. Elevated pH can increase the activity of fecal proteolytic enzymes that can degrade the extracellular matrix of the skin and eventually destroy tissues [61].

Skin pH is believed to be one of possible factors that promote candida infection, which diabetics are particularly prone to suffer from. Dimorphism of *Candida albicans* exists, as the blastospore form is related to acidic pH and the mycelial form is associated with alkaline pH [62,63]. The mycelial form of *Candida albicans* has been verified in clinical studies via its pathogenicity [64,65,66]. Gil Yosopovitch and coworkers [67] found that diabetic patients had significantly higher skin pH than that of normal control skin. Moreover, the skin pH values of diabetic patients with candidiasis was higher than that of diabetic patients without candidiasis. This suggests a possible correlation between skin pH and candida infection.

### 2.4. Skin Care

There are several structural differences between the skin of newborns and that of adults. Because newborn skin has a poorly functional skin barrier, and because newborns have fragile skin that can be easily damaged, proper and adequate steps must be taken to provide protection. S. Dhar [68] suggested that newborn bathing should not last longer than 5 min, or it could increase the hydration of the skin and decrease the friction threshold. Moreover, soaps and cleansers should be minimally used in the first few weeks of life for newborns, because any attempt intended to raise the skin pH would promote the number of bacteria and increase the TEWL [69]. There is not enough evidence that soaps have long-term impact on infants, but in the short term, it could disturb their acid mantles [31,33,34]. Synthetic detergents such as cocoyl isethionate, and sodium lauryl sulphate are soap substitutes that have pH values similar to skin and are more moderate than soaps. They do not alter skin pH and microflora, but they disintegrate rapidly and can cause skin dryness [70].

The spread of lesions in infantile seborrheic dermatitis can be limited by mineral oil or vegetable oil and minimizing the use of shampoos for cleaning the hair. Sarkar et al. [71] recommended the use of coconut oil as an emollient for application to neonatal skin due to its easy availability and economy. Skin irritation can also be avoided by using fragrance-free baby products.

For adult skin care, Y.C. Jung and coworkers [72] suggested that low skin pH could be maintained by increasing hydration, the presence of low skin pH induced higher sebum excretion rate, and the combined effects could suppress skin aging by reducing wrinkle length and depth. In work by S. H. Youn et al. [73], they found it helpful to control acne in the T-zone by increasing skin pH to 5.5–6 for female acne patients, and by decreasing skin pH to approximately 5.5 for male patients. Saba et al. [74] recommended that the ideal pH for body wash, soap, or cleanser would be in the range of 4.5–6.5, which is similar to that of the skin’s acid mantle. Moreover, syndets, synthetic detergents, are less irritating and preferred. Nix and coworkers [75] suggested that the selection of skin cleaning products should take multiple factors into consideration, namely formulation, ingredients, skin compatibility, pH, and related infection control issues. Skin cleaning products with a pH value higher than 7 can disrupt skin barrier function. It is particularly important for elderly patients to select products with a pH value of 4–7 because the skin of the elderly is dryer, more prone to cracking, and recovers more slowly from damage caused by alkalinity.

## 3. Wounds

Wounds are divided into two categories, acute and chronic. Acute wounds can be healed predictably with normal wound healing approaches, and chronic wounds develop following one or more failures in the normal wound healing process. The wound healing process has four phases and has been described in detail above.

Acute wounds, sudden injuries to the skin, vary from superficial scratches to deep skin damages that can happen anywhere on the body. The causes of acute wounds vary but primarily include abrasion, puncture, laceration, and incision. Classified by causality, the two dominant types of acute wounds are surgical and traumatic. Surgical wounds are intentionally created for medical reasons, and traumatic wounds are randomly caused by external force. Severe pain is associated with wounds, which are especially prevalent among patients with fragile skin.

Chronic wounds remain in an inflamed state that delays healing for long periods lasting several months. Chronic wounds may take years to recover or may never heal, leading to physical and psychological suffering as well as considerable pressure on the social healthcare system. Diabetic patients with chronic wounds are at especially high risk for bacterial infections, because the slightly alkaline pH of chronic wounds promotes bacterial colonization. Figure 3a shows the pH values of acute wounds and chronic wounds in Figure 3b (adapted from [10]). During healing, the pH of acute wounds gradually declines to a level approximating that of the acid mantle. Chronic wounds, however, remain slightly alkaline. Monitoring and manipulating pH can be a critical tool for preventing and treating chronic wounds. Studies have reported on pH value monitoring combined with pH-activated drug control release systems and smart wound dressings and bandages [11,12,13,14,15].

Severe chronic wounds with bacterial infections can evolve into bacteremia, sepsis, or septicemia and severe deterioration in wound status. Severely chronic wounds can result in amputation, a frequent occurrence with diabetic foot ulcers, and a major issue in modern healthcare [76,77]. Diabetic patient wound care management must be undertaken with care and rigor. In 2014, chronic wounds impacted 15% of all Medicare beneficiaries in the United States, with an estimated cost of $28–$32 billion annually [78]. Among the 15% of Medicare beneficiaries (8.2 million in population), who had at least one type of wound or infection, surgical infections were most prevalent (4.0%), followed by diabetic infections (3.4%) [79]. The huge cost of nonhealing chronic wounds has become an urgent issue that should be taken seriously.

### 3.1. The Role of pH Value in Wound Healing

pH value is one of the critical factors involved in the wound healing process for both acute wounds and chronic wounds; it affects matrix metalloproteinase activity, fibroblast activity, keratinocyte proliferation, microbial proliferation, biofilm formation, and immunological responses [80]. Different stages of the wound healing process may require environments of different pH to recover from skin damage and infection. Imbalances in pH can result in serious chronic wounds.

Matrix metalloproteinases (MMPs) are a family of more than 20 proteases. They are able to degrade extracellular components and facilitate the removal of damaged tissues after new tissues have formed. However, excessive protease amounts can interrupt the wound healing process, since they cause endothelial cell damage and degrade the epidermal–dermal junction, which eventually destroys the newly forming tissues [9,80]. To limit expression of MMPs, metalloproteinases (TIMPs) act as inhibitors; they are often at low levels in chronic wounds. Balancing the levels of MMPs and TIMPs is necessary for successful wound healing. Figure 4 (adapted from [9]) depicts the pH-dependent activity of four proteases important in wound healing. Because chronic wounds remain in an alkaline milieu, this could be one cause for high expression of MMPs that lead to unrecovered chronic wounds. However, most proteases can be inactivated in an acid milieu, so decreasing the pH value in wound sites may be an effective method for treating chronic wounds.

Lowering the pH of the wound environment may enhance wound contraction. Pipelzadeh et al. [8] stimulated the responses of strips of rat superficial fascia in vitro using physiological solutions at different pH values (5.5, 6.6, 7.3, 8.1) and containing adenosine, calcium and potassium ions, and mepyramine. Their results suggest that there was an at least sixfold increase in the responsiveness to adenosine under acidic conditions (pH 5.5) compared with controls, while in alkaline conditions, responses were not significantly different from control responses.

Lönnqvist et al. [81] found that human keratinocytes cultured in medium with a pH value of 6.0 showed a decreased ability to re-populate the scratched area compared to controls, and no wounds presented re-epithelialization when cultured at pH 5.0. These results suggest that any efforts to alter local wound pH should remain over pH 5.0, which could ensure better re-epithelialization.

It is possible that different stages of wound healing require environments of different pH. An acid milieu could improve fibroblast proliferation [82], whereas neutral and alkaline environments with a pH range of 7.21–8.34 are better for re-epithelialization [83,84]. To prevent the development of chronic wounds, an acid milieu is useful for depressing excessive amounts of MMPs and helps to form the protective acid mantle and normal skin barrier.

### 3.2. pH in Infected Wounds

Breidenbach et al. [85] showed that tissue microbial levels equal to or higher than 10^4^ cfu/g were responsible for delayed wound healing. However, bacterial biofilm can affect the condition of chronic wounds, alter host immune response, affect interactions with other microorganisms, and alter pH, temperature, and nutrient levels in wound regions [86].

Ammonia, a by-product of bacterial metabolism, raises tissue pH [87]. Human pathogenic bacteria grow well when the pH value is above 6, which could lead to more complicated situation for chronic wounds. Because the pH fluctuation in chronic wounds (Figure 3b) is significantly distinct from that of the normal acid mantle, the skin fails to prevent bacterial colonization. Furthermore, the activity of MMPs is heightened when the milieu changes from neutral to alkaline, and infected chronic wounds are subjected to repeated inflammatory states that prevent healing. Changes in pH change can also influence the toxicity of bacterial end products and thus affect their enzyme activity. For instance, lowering pH has been shown to induce structural changes in staphylococcal enterotoxin (SEC2), which has been discussed above [51]. Decreasing pH in wound regions may not only suppress excess MMP expression but also inhibit bacterial growth and toxicity, making it a synergistical method for treating chronic wounds.

Percival et al. [88] reviewed the effect of some antiseptics on pH. In other research, Percival and coworkers [89] found that the antimicrobial performance of silver dressing was significantly enhanced at a pH of 5.5 when compared with a pH of 7.0. It was presumed that the oxidized form of silver present in an acidic environment was active for inhibiting bacterial growth. Molecular iodine (I_2_) with its active forms (I_2_, H_2_OI^+^) in solution is calculated to be in the pH range of 3–9 and is thought to have great antimicrobial activity. The greatest antimicrobial effect is found within an acid milieu having a pH between 3 and 6, since H_2_OI^+^ is the critical biocidal agent most released in this environment [90]. Polyhexamethylene biguanide (PHMB) is a cationic antimicrobial that can inactivate the efflux pump of bacterial cell membranes. The maximal antibacterial effect of PHMB has been shown to occur at a pH range of 5–6 [91]. Chlorhexidine (CHX) is a biguanide cationic detergent similar to PHMB and is more stable at a pH range from 5 to 8 in solution, but its antimicrobial efficacy is optimal in a range of pH 5.5–7.0, unlike PHMB, which is more effective in an acidic milieu [92]. There are a number of other examples not cited in this review, but overall, optimizing pH values can benefit wound healing and inhibit microbial growth.

### 3.3. Wound Dressings

Wound dressings are barriers that prevent wounds from further mechanical damage or infection. An ideal wound dressing should not only serve as protection but should also be able to absorb wound exudates, maintain a wet environment around the wound, and allow normal transportation of nutrients and gases. There are various types of wound dressings already available in clinical use and on the market, such as plasters, strips, tapes, foams, beads, occlusive films, hydrocolloids, hydrogels, alginate, charcoal dressings, silicone gel, etc.

In recent years, some studies have focused on developing smart wound dressings or bandages different from conventional wound dressings and featuring additional functions, such as radiation-activated drug control-released system, the detection of wound status, and wireless connections between smart wound dressings and mobile phones. Smart wound dressings equipped with pH and temperature sensors can utilize the pH change in wounds to activate drug release for monitoring and treating proposes. Mirani et al. [11] designed a multifunctional hydrogel-based wound dressing capable of colorimetric measurement of pH, indication of bacterial infection, and the release of antibiotic agents at the wound site. They also developed an image analysis application that can record and analyze digital images captured by a smartphone. This facilitates monitoring of treatment strategies that patients can manage at home. In 2018, Mostafalu and coworkers [13] reported a smart bandage for treating and monitoring chronic wounds. This hydrogel-based dressing is loaded with a temperature sensor, a pH sensor, a flexible heater, and a thermo-responsive drug release system. It may be well suited for the treatment of chronic wounds. Moreover, the sensor can be connected to a mobile phone via Bluetooth for visual recording of wound status that allows patients to self-manage wounds. Liu and coworkers [14] developed a smart hydrogel wound patch incorporating modified pH indicator dyes. It showed optimal mechanical properties under different calcium and water concentrations. When pH increased, the color of the hydrogel patches underwent a color transition from yellow to orange and red, which matched the clinically meaningful pH range of chronic or infected wounds. Kiaee and coworkers [15] reported on an electronic wound dressing with active topical drug delivery in response to electrically induced pH change. In basic environments, this dressing released drugs in response to the dehydration process. With this smart dressing, chronic wounds can be systematically and effectively treated.

### 3.4. Wound Care

Debridement, transplant, negative-pressure wound therapy (NPWT or VAC), and antibiotic ointments are commonly used for wound management. Debridement literally means the removal of dead or infected tissues to improve wound healing. There are several techniques to remove damaged tissues including surgical, mechanical, chemical, autolytic, and maggot-based debridement. Skin transplantation is successfully used to close open chronic wounds by placing autologous grafts or allogeneic grafts on the wound surface. This treatment option can support the rate of epithelialization of the wound areas, and rapid covering of central wound areas with keratinocytes [93]. To investigate the effect of NPWT, Armstrong and coworkers [94] analyzed 162 patients (77 assigned to NPWT and 85 control) in a randomized clinical trial and found that more patients healed in the NPWT group. The speed of wound closure and tissue formation was faster in the NPWT group than in control groups. NPWT seems to be safe and effective for treating complex diabetic foot wounds and could potentially reduce re-amputation cases compared to standard care. Antibiotic ointments are widely used for preventing wound infections. It is a fairly common wound management practice in clinic and personal wound care. However, multidrug-resistant (MDR) pathogens have rapidly emerged due to the abuse of antibiotics and have created a growing a threat to human health.

Bowler and coworkers [95] reviewed infections in various wounds and suggested some associated approaches for wound management. Wound cleansing and surgical debridement are complementary with antimicrobial therapy because they could help reduce the opportunity for infection by reducing microbial load and provide better antibiotic penetration. Moreover, debridement can also avoid blood clots accumulating in tissue debris and consequently reduce the potential for microbial growth. To address the issue of MDR pathogens, nanoparticles have been applied to wound dressings to prevent infection. There are already several review articles discussing the effects of silver and silver nanoparticles in wound management [96,97,98,99]. However, because silver is toxic towards mammalian cells, therapeutic window questions remain unanswered. Svitlana Chernousova and Matthias Epple [100] reviewed the toxicity of silver (silver ions and silver nanoparticles) in bacterial, cellular, and animal tests, and examined its impact on the environment. Silver toxicity is associated with particle size, size distribution, morphology, crystallinity, surface functionalization, charge, dose, and concentration. For these reasons, silver should be critically assessed and well-characterized before being used in consumer products, cosmetics, and medical products. Additionally, there is an emerging concern regarding silver-resistant bacteria [101].

Infants have fragile skin and poor skin barriers that result in easy damage. For these reasons, wound care for infants should be particularly careful and rigorous. Strodtbeck and coworkers [102] suggest several strategies for infant wound care that are shown in Figure 5 (adapted from [102]). The scientific rationales for each care strategy are further explained in Strodtbeck’s article. Stahl and coworkers [103] reported that cleansing and topical antibacterial use has helped prevent infant open wounds from getting infected from synthetic heterografts, septal patches, valves, conduits, or homograft outflow tracts used during the reconstruction of thoracic wounds. Those prostheses could increase wound complications and the risk of endovascular infection. Bookout and coworkers [104] reported on successful treatment of an infant, a toddler, and an adolescent NPWT. Notably, the infant’s labial/buttock wound was completely closed without skin transplantation. They also suggested that adjustments of NPWT should be relative to age, size, and tolerance to therapy.

We suggest monitoring pH change during wound management as a means of evaluating wound healing. Measuring pH value in wound sites is relatively simple and fast. It could be used to provide a rough reference for physicians to diagnose and assess wound healing state, including whether wounds are receiving effective treatment or are becoming chronic, unhealing wounds. Additional examination and analysis, including ultrasound, wound depth measurement, and infection testing could supplement this initial, rough assessment to best determine treatment protocol.

## 4. Future

pH value is one of the most important factors affecting physiological performance. It may be very useful for making diagnoses and for monitoring and treating wounds and skin-related diseases. pH influences skin barrier functions and plays a critical role in regulating the activation of proteases related to wound healing processes and the development of chronic wounds. Furthermore, it has been correlated with bacterial enterotoxin activity and antibacterial efficacy. Monitoring pH changes in wound regions is a fast and simple method for monitoring wound healing processes and preventing the formation of chronic wounds. Developing a smart bandage equipped with a pH sensor or other systems (e.g., drug control-released carriers) have become increasingly popular in recent years. Besides, tools for non-invasive diagnosis and therapy monitoring in dermatology (e.g., raster-scan optoacoustic mesoscopy (RSOM) and in vivo reflectance confocal microscopy (IVCM)) have also been introduced [105,106]. They are expected to promote additional tools and processes to promote wound healing. While pH value is a simple physiological concept, it could promote a significantly positive impact on not only clinically relevant diagnoses but personal health management in the home.

## Figures and Tables

**Figure 1 diagnostics-10-00107-f001:**
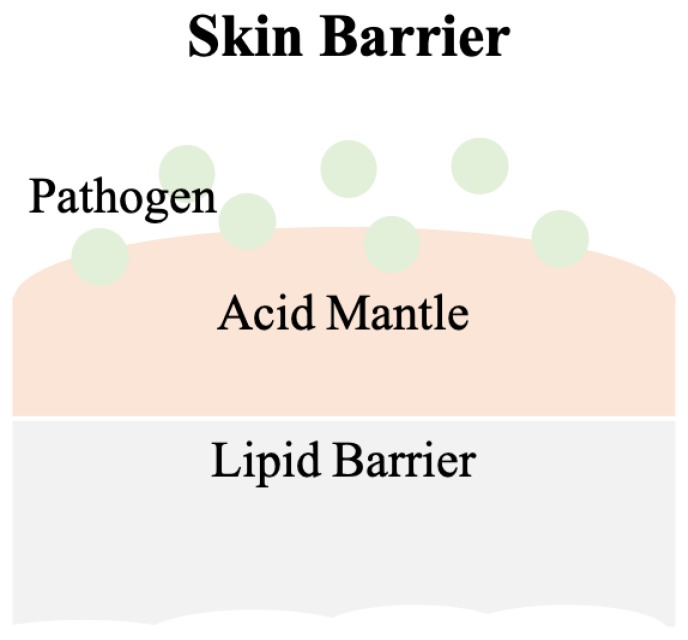
The scheme of the skin barrier. Lipid barrier sits atop the acid mantle and displays a pH value of 4–6. Pathogens can be various bacteria. Slight acidic environment is a disadvantage for bacterial colonization.

**Figure 2 diagnostics-10-00107-f002:**
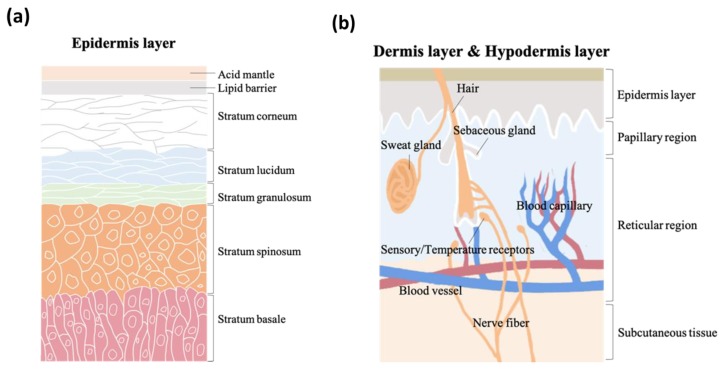
The schematic structure of (**a**) epidermis layer, (**b**) dermis layer, and hypodermis layer.

**Figure 3 diagnostics-10-00107-f003:**
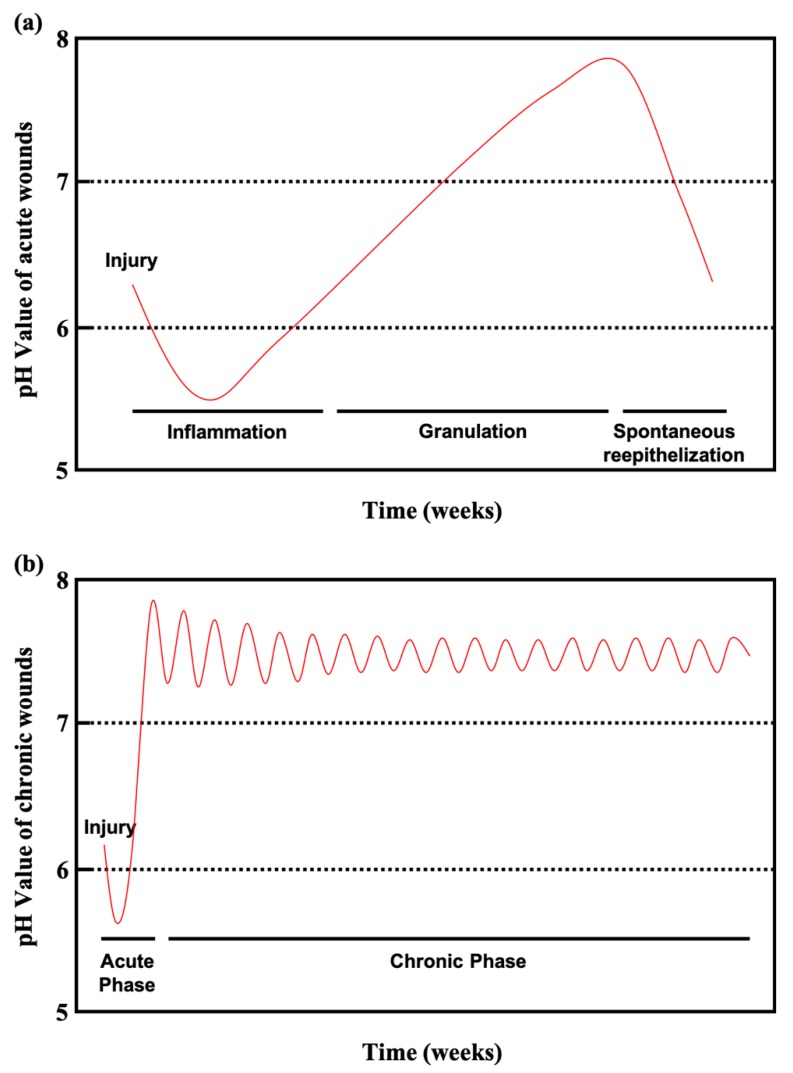
(**a**) Course of the pH milieu in acute wounds. (**b**) Course of the pH milieu in chronic wounds (adapted from [10]).

**Figure 4 diagnostics-10-00107-f004:**
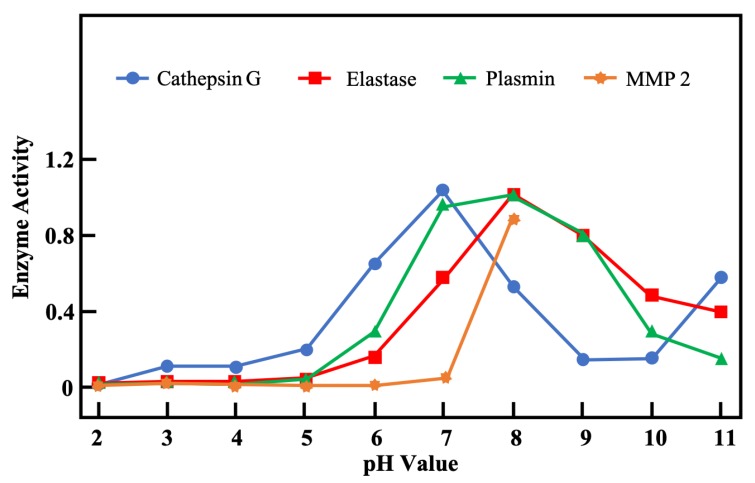
Assessment of pH-dependent activity profiles of four proteases important in wound healing (adapted from [9]).

**Figure 5 diagnostics-10-00107-f005:**
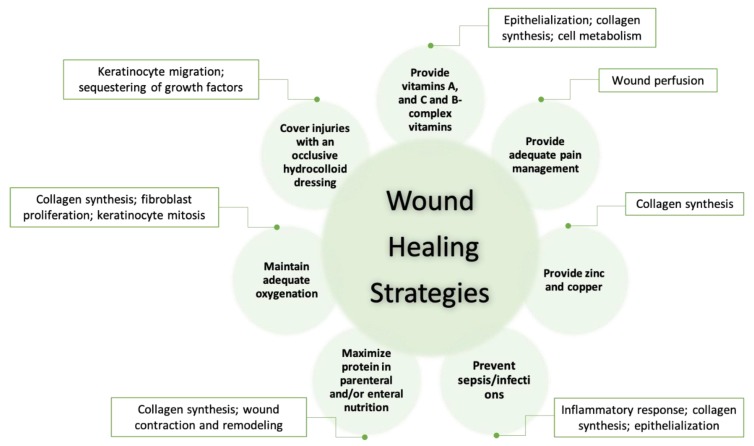
Strategies to promote wound healing in newborns/infants (adapted from [102]).

**Table 1 diagnostics-10-00107-t001:** Mean values (MV ± SD) arranged by age groups and localization (adapted from [26]).

	**AG I**		**AG II**		**AG III**		**AG IV**		**AG V**		**Mean**	
	**♀**	**♂**	**♀**	**♂**	**♀**	**♂**	**♀**	**♂**	**♀**	**♂**	**♀**	**♂**
**F**	5.01 ± 0.42	4.31 ± 0.28	4.98 ± 0.50	4.47 ± 0.33	4.92 ± 0.40	4.50 ± 0.36	5.10 ± 0.49	4.56 ± 0.30	4.67 ± 0.32	4.57 ± 0.35	4.94 ± 0.45	4.48 ± 0.33
**C**	5.33 ± 0.32	4.66 ± 0.38	5.27 ± 0.44	4.77 ± 0.33	5.19 ± 0.29	4.74 ± 0.36	5.35 ± 0.46	4.82 ± 0.27	5.21 ± 0.41	4.93 ± 0.32	5.27 ± 0.39	4.78 ± 0.34
**N**	5.09 ± 0.32	4.62 ± 0.37	5.09 ± 0.36	4.72 ± 0.36	5.08 ± 0.34	4.63 ± 0.34	5.37 ± 0.49	4.76 ± 0.28	5.05 ± 0.41	4.73 ± 0.29	5.13 ± 0.40	4.69 ± 0.33
**A**	5.12 ± 0.31	4.49 ± 0.42	5.22 ± 0.40	4.50 ± 0.36	5.17 ± 0.38	4.53 ± 0.30	5.44 ± 0.45	4.62 ± 0.37	5.12 ± 0.43	4.58 ± 0.37	5.21 ± 0.41	4.54 ± 0.36
**H**	5.10 ± 0.37	4.33 ± 0.36	5.19 ± 0.40	4.41 ± 0.32	5.03 ± 0.35	4.38 ± 0.31	5.25 ± 0.38	4.55 ± 0.42	4.88 ± 0.41	4.46 ± 0.40	5.09 ± 0.40	4.42 ± 0.37
	**Female (Age ± SD)**						**Male (Age ± SD)**					
**AG I**	24.73 ± 2.59						25.70 ± 2.48					
**AG II**	33.30 ± 2.92						33.23 ± 2.90					
**AG III**	43.63 ± 2.81						44.23 ± 3.09					
**AG IV**	55.17 ± 2.84						54.20 ± 3.13					
**AG V**	66.57 ± 4.31						66.63 ± 2.61					

F = Forehead; C = Cheek; N = Neck; A = Forearm; H = Hand.

**Table 2 diagnostics-10-00107-t002:** Factors that have influences on skin pH.

Factors	Influences	References
Age	Increased age is associated with skin pH values that are in steady fluctuation. This contributes to difficulties in making comparisons. Females in [26] show significantly higher skin pH values than males.	[23,26,41]
Ethnic	Whites have slightly higher pH than blacks, but there are mostly no statistically significant differences between ethnic groups.	[42,43,44]
Sebum	There is a negative correlation between sebum secretion and skin pH that is statistically significant in only the skin U-zone with acne and in the skin T-zone without acne, yet the correlation is not strong enough to explain the impact of sebum secretion on skin pH.	[45]
Sweat	Sweat is predominantly composed of sodium chloride, lactic acid, urea, and fatty acid. The acidic nature of human skin is partially due to lactic acid in sweat. As the rate of eccrine sweat increases, the concentrations of ammonia, pyruvic acid and lactate decrease, hence the pH increases.	[28,29,30,46,47]
Soaps and detergents	The use of soaps, synthetic detergents and even tap water will raise skin pH in a short-term effect.	[27,31,32,33,34,35]
Cosmetic products	Skin pH values drop after the application of cosmetic products, which can buffer the impact of tap water washing.	[36,37]
Occlusive dressings	Under occlusion, skin pH increases, but after removing occlusive dressings, the pH milieu drops back to an acidic nature. The application of occlusion has a short period effect on skin pH values.	[38,39,40]

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
