# Peer review of "Role of pH Value in Clinically Relevant Diagnosis"

_diagnostics, 2020, doi:10.3390/diagnostics10020107_

Round 1

Reviewer 1 Report

This was a really interesting and relevant review of the role of pH in clinically relevant diagnosis, in particular in relation to skin structure and wound status.  It may be of benefit for others searching for your article to include skin and/or wounds in your title.

Significance

This article provides a great, up-to-date review of pH in this area and will be of relevance to clinicians and researchers in the area of skin and wound care.

Quality of presentation

A really well written and structured paper which was easy to read.  Two very small typos

Line 348 investicate

Line 394 concept written twice

Scientific Soundness

A well put together literature review. 

It was difficult for me to review references as titles of articles were not included in reference list – this is not a form of referencing that I am familiar with so will leave with editors as to whether this is usual practice.  I also thought there was information in-text that should have been referenced i.e. 2. Skin – has no references and a number of other statements such as ‘Severe pain is associated with wounds, which are especially prevalent among patients with fragile skin’, I would have referenced but again I will leave it with the editor to make a call on whether this is appropriate referencing for this review type of journal article as the majority of referencing is well done.

I’m unsure of the relevance of 3.4 Wound Care as this does not mention pH.

Overall merit

A great review which I look forward to seeing published.

Reviewer 2 Report

Thank you very much for the opportunity to provide review comments on this valuable manuscript regarding the role of pH value in skin and wound healing.

Major comments

In section 3.4. Wound care, it is better to focus on the role of pH value. This section is too broad to follow.

Minor comments

Since this review article is dedicated to representing the role of pH value not only in wound healing but also in the skin, it is better to include this purpose in the abstract. Table 1: you need to add the explantation regarding AGI-V. Table 2: Be consistent in present or past tense. Some parts are maybe from the one that might have been submitted to other journals before this submission (eg., Table III??). You must delete those unnecessary data. Regarding the description by referencing Elizabeth's study, you need to connect to pH value. "Last year" should read in 2018.
